

# A DoS attack detection method based on adversarial neural network

Yang Li and Haiyan Wu

Zhengzhou Police University, Zhengzhou, Henan, China

## ABSTRACT

In order to analyze the influence of deep learning model on detecting denial-of-service (DoS) attacks, this article first examines the concepts and attack strategies of DoS assaults before looking into the present detection methodologies for DoS attacks. A distributed DoS attack detection system based on deep learning is established in response to the investigation's limitations. This system can quickly and accurately identify the traffic of distributed DoS attacks in the network that needs to be detected and then promptly send an alarm signal to the system. Then, a model called the Improved Conditional Wasserstein Generative Adversarial Network with Inverter (ICWGANInverter) is proposed in response to the characteristics of incomplete network traffic in DoS attacks. This model automatically learns the advanced abstract information of the original data and then employs the method of reconstruction error to identify the best classification label. It is then tested on the intrusion detection dataset NSL-KDD. The findings demonstrate that the mean square error of continuous feature reconstruction in the sub-datasets KDDTest+ and KDDTest-21 steadily increases as the noise factor increases. All of the receiver operating characteristic (ROC) curves are shown at the top of the diagonal, and the overall area under the ROC curve (AUC) values of the macro-average and micro-average are above 0.8, which demonstrates that the ICWGANInverter model has excellent detection performance in both single category attack detection and overall attack detection. This model has a greater detection accuracy than other models, reaching 87.79%. This demonstrates that the approach suggested in this article offers higher benefits for detecting DoS attacks.

## INTRODUCTION

Attack detection, which serves as the cornerstone of attack defense, is crucial for preventing security threats and safeguarding network resources from attacks (*Zeng et al., 2022*; *Syed et al., 2020*). Enterprise product security discussions must inevitably touch on how to prevent denial-of-service (DoS) attacks, particularly given the prevalent micro-service design (*Adefemi Alimi et al., 2022*; *Wang & Li, 2021*; *Latah & Toker, 2020*). Networked control systems have been implemented in industrial process control, unmanned aerial vehicle (UAV) control, and other disciplines as a result of the rising status of network communication technology in the field of industrial control (*Ortega-Fernandez & Liberati, 2023*). Although the addition of network links increases the control system's flexibility

Corresponding author
Yang Li, liyang@rpc.edu.cn

and convenience, it also poses security problems due to internal unreliability and external security threats (*de Neira, Kantarci & Nogueira, 2023*; *Abreu Maranhão et al., 2020*). Wireless network data packets can be hampered, intercepted, or even altered as they are being transmitted. The most frequent type of these malicious attacks on network systems is a DoS attack (*Sokkalingam & Ramakrishnan, 2022*; *Alashhab et al., 2022*; *Singh & Jayakumar, 2022*). DoS attacks make it impossible for data packets sent over the network link to be delivered normally, which delays or even causes the control system to operate in open-loop mode. Such severe repercussions are intolerable for the network system (*Vedula et al., 2021*; *Tao et al., 2022*; *Almaraz-Rivera, Perez-Diaz & Cantoral-Ceballos, 2022*).

The goal of a DoS attack is to prevent authorized users from having regular access to shared services or resources. This can cause system overload and, to variable degrees, prohibit authorized requests from receiving timely responses (*Kumar et al., 2021*). The strong real-time limitations of many networked control systems will be destroyed, even by a brief DoS attack (*Manickam et al., 2022*). Additionally, as the dynamics between attack types and defense systems have changed, new, complex, and integrated DoS attack techniques have emerged, while conventional detection techniques have not lived up to expectations (*Tinubu et al., 2022*; *Khan, 2021*; *Li et al., 2022*). Typical protective strategies include screening false IP addresses and restricting unused ports and services on the router. The most sophisticated picture classification model available today is the convolutional neural network (CNN) model in machine learning. It is a promising potential technology to categorize network traffic and identify malicious traffic (*Aladaileh et al., 2020*). However, when it comes to identifying unknown threats, even this machine learning approach has a high rate of false alarms and poor detection precision. To ensure the secure operation of networked control systems, it is important to research fresh and potent detection techniques for DoS attacks.

As a result, this article initially examines the fundamentals and attack tactics of DoS attacks before looking into the most recent DoS attack detection techniques. A distributed DoS attack detection system based on deep learning is ingeniously established in response to the investigation's limitations. This system can quickly and accurately identify the traffic of distributed DoS attacks in the network that needs to be detected and then promptly send an alarm signal to the system. The incomplete network traffic characteristics in DoS attacks are then addressed by proposing a model named the Improved Conditional Wasserstein Generative Predictive Network with Inverter (ICWGANInverter). It automatically learns the advanced abstract information of the original data and uses the reconstruction error method to determine the best classification label. This approach offers a fresh perspective on the study of DoS attacks.

This article has made remarkable innovations in the field of intrusion detection, aiming at improving the system performance, intelligence and robustness, paying special attention to the robustness in the face of zero-day attacks. Firstly, the ICWGANInverter model is introduced, and its experiments on NLS-KDD datasets show excellent feature reconstruction and classification performance under different noise factors. Secondly, by comparing the comprehensive performance of KDDTest+ and KDDTest-21 datasets, it is proved that the ICWGANInverter model has obvious advantages over other advanced

models in many performance indexes, especially in DoS attack detection. Different from the traditional intrusion detection methods, this article not only focuses on improving the accuracy of detection, but also focuses on the processing of a lot of noise. Through experiments on NLS-KDD datasets, the ICWGANInverter model performs well in feature reconstruction and classification performance test under different noise factors. The results show that the feature reconstruction ability of the model is inversely proportional to the noise intensity, and it is highly robust to different noises. This is an innovation of this study in dealing with the noise problem that is common in the actual network environment. In addition, by comparing with the research of related scholars, this article fully draws on the exploration of SVM classifier, deep learning technology and multi-layer perceptron strategy by different scholars, showing the integration and innovation of various technologies. This comprehensive method makes this article have great innovation value on many levels, not only paying attention to the performance improvement of traditional methods, but also making outstanding contributions to dealing with actual network noise and complex environment.

## LITERATURE REVIEW

The core of a DoS attack is to utilize flaws in the present network protocol to indiscriminately drain the resources of the target host in a variety of ways, preventing it from providing regular services and possibly even crashing or going down. Several academics have studied the defensive and detection tactics for this type of attack as a result of the serious repercussions of this attack. To evaluate the indirect impact of misuse of supervision on the assault propagation effect, *Siyal et al. (2021)* designed and empirically tested a mediation model. Data transmission depended heavily on smart network security, as *Chen, Wawrzynski & Lv (2021)* explained. A deep learning-based Distributed Denial of Service (DDoS) assault intrusion detection system based on three models, CNN, deep neural network (DNN), and circular neural network, was proposed by *Ferrag et al., (2021)*. The effectiveness of each model in two classification kinds is examined using two fresh real traffic datasets (binary and multi-class). To detect these sophisticated attacks with unexpected patterns, *Kim et al. (2020)* constructed a model based on CNN and compared its performance with that of a recurrent neural network (RNN). They developed an intrusion model based on deep learning, giving particular attention to DoS attacks.

*Mohammed, Rashidi & Salih (2022)* developed a novel approach based on software networking and cooperative learning to enhance the security of the Internet of Things (IoT). According to the suggested solution, the network domain was divided into numerous sub-domains, each of which had its own controller for exchanging security rules with other sub-domains. By utilizing the special feature of software-defined network architecture and employing the exponential weighted moving average protection mechanism in statistical distance, *Ghasabi & Deypir (2021)* proposed a statistical approach to identify and mitigate distributed DoS assaults. The findings demonstrated how rapidly this technique can identify attack traffic and apply countermeasures. A brand-new technique to identify DoS assaults in a cloud computing context was proposed by *Kushwah & Ranga (2020)*. The proposed

system's accuracy for identifying attacks on NSL-KDD datasets was 99.18% and it was developed using a voting limit learning machine. *Pujol-Perich et al. (2022)* put forward a graph neural network (GNN) model specially for processing and learning. By paying attention to the structural pattern of attacks, especially the relationship among streams, the robustness of Network Intrusion Detection Systems (NIDS) can be improved (*Pujol-Perich et al., 2022*). *Xiao et al. (2023)* put forward a control area network graph attention networks (CAN-GAT) model, which was used to realize the anomaly detection of in-vehicle networks. (*Deng et al. (2022)* proposed a collaborative deep reinforcement learning (CDRL) model based on graph neural network to generate resource allocation and mitigation strategies. *Zhang et al. (2021)* emphasized the opportunities of machine learning and deep learning, especially the effective algorithm of deep learning in extracting useful information from training data.

While there are numerous ways to identify DoS assaults at the moment, progress has been made. However, there is still space for growth in terms of detection effectiveness, packet loss rate, and false alarm rate. Due to the game theory between its generator and discriminator, the GAN is appropriate for the study of assault and defense. Consequently, by employing the optimal GAN model to identify DoS attacks, this article can offer a novel solution for this type of network attacks.

## WORKFLOW

In this article, the basic knowledge and attack strategy of distributed denial of service (DoS) attack are deeply studied, which provides readers with a comprehensive understanding of this field. Then, in view of some limitations of the existing research, this article skilfully establishes a distributed DoS attack detection system based on deep learning. This system has the ability to quickly and accurately identify the distributed DoS attack traffic in the network to be detected, and can quickly send an alarm signal to the system. This system provides an efficient means to deal with distributed DoS attacks for network security. Then, in order to solve the problem of incomplete network traffic in DoS attacks, this article proposes a model called ICWGANInverter. This model automatically learns the high-level abstract information of the original data, and uses the reconstruction error method to determine the best classification label to better deal with the network traffic problem in DoS attacks. Finally, the ICWGANInverter model is used to detect continuous incomplete feature attacks and the performance of different attack detection models is compared. Figure 1 shows the research workflow.

## RESEARCH METHODOLOGY

### DoS attack and attack strategy

DoS attacks usually start a massive attack on the target system by taking advantage of weaknesses in systems, services, and transmission protocols. They also use system resources and bandwidth by sending large amounts of data packets that are larger than the target computer's processing capacity, or they result in program buffer overflow errors, which

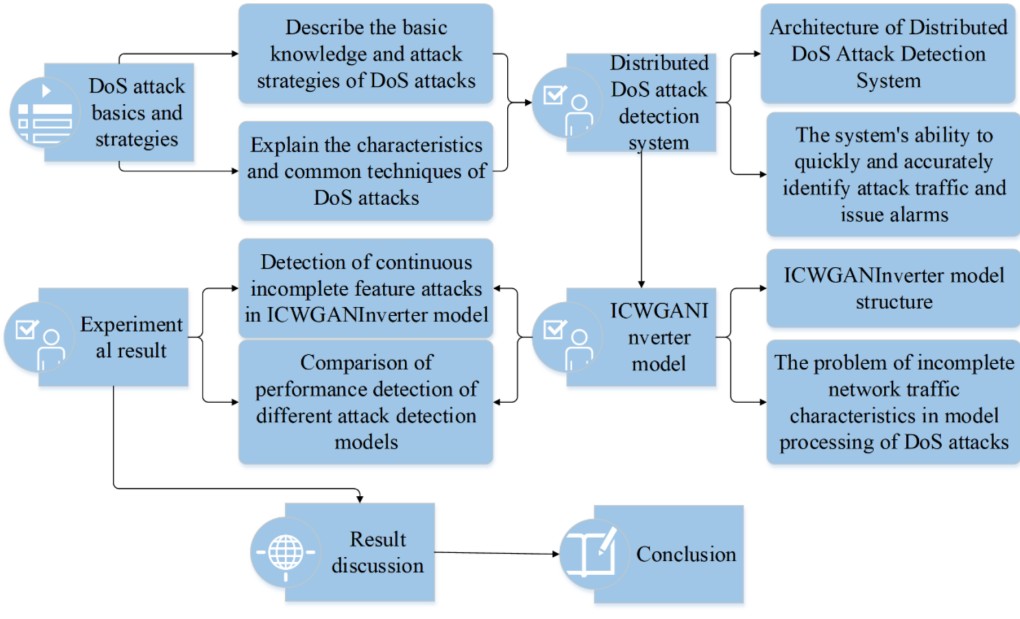

**Figure 1 Research workflow.**

prevent the target system from processing normal requests from legitimate users or from providing normal services, thus paralyzing network services.

Attacks against networked control systems can be broadly categorized into three categories: DoS attacks, assaults involving inaccurate data, and replay attacks. DoS attacks are the most typical of them. DoS attacks cause packet loss in the system by flooding important nodes in the communication network with a high number of pointless packets that prevent the receiver from responding to legitimate packets in time. Attackers can also increase the bit error rate of the data packet by interfering with the channel, which will cause the original data packet to be deleted owing to transmission problems and decrease the signal-to-interference plus noise ratio (SINR) of the network control system.

There are three types of DoS assaults that are frequently used: disseminated, IP spoofing, and direct. (1) A direct attack occurs when a significant number of falsified Transmission Control Protocol (TCP) packets are sent directly by the attacker. Because the attacker in this type of assault does not mask their IP address, the attack originates from a single source, making it simple to identify and stop the attacker. (2) IP spoofing assaults involve attackers fabricating IP addresses, which makes it harder for defenders to track down their targets. IP address forging technology can be implemented in a simple manner. Prior to writing the message to the output device and sending it to the target host, a structure with an IP message format is first formed, followed by the filling in of the structure's source address with a fictitious IP address. DDoS: It is nearly impossible to pinpoint the origin of an attack when the attacker is in possession of a large number of infected machines and uses botnets. Some attackers may even instruct each device to fake its IP address, combining the

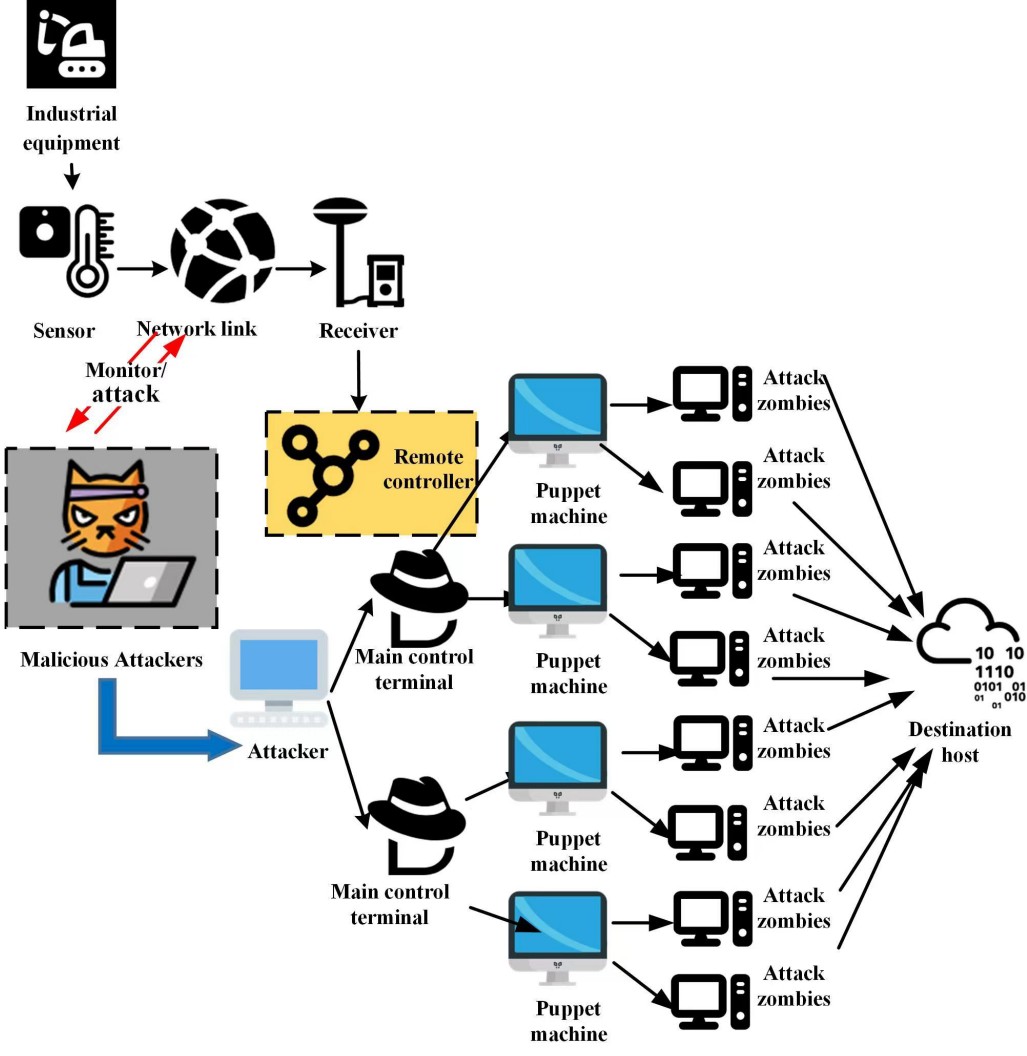

**Figure 2** **DDoS attack schematic diagram.** Source credits: Industrial equipment, https://yesicon.app/ healthicons/machinery-negative, (c) 2021 Resolve to Save Lives; Sensor, https://yesicon.app/cbi/motion- sensor-temperature, CC BY-NC-SA 4.0; Network link, https://yesicon.app/entypo/network, CC BY-SA 4.0; Receiver, https://yesicon.app/gis/gnss-antenna, CC BY 4.0; Malicious attackers, https://yesicon.app/ openmoji/hacker-cat, CC BY-SA 4.0; Remote control, https://yesicon.app/bx/network-chart, CC BY 4.0; Attackers (computers) https://yesicon.app/twemoji/desktop-computer, CC BY 4.0; Main control termi- nal (hat) https://yesicon.app/simple-icons/cyberdefenders, CC0 1.0; Puppet machine (Mac screen) https: //yesicon.app/emojione-v1/laptop-computer, CC BY-SA 4.0; Attack zombies (desktop settings) https:// yesicon.app/fa6-solid/computer, CC BY 4.0; Destination host (cloud) https://yesicon.app/simple-icons/ tryhackme, CC0 1.0.

two attack strategies and doing irreparable harm to the victim. The DDoS attack principle is depicted in Fig. 2.

Figure 2 illustrates the interaction and relationships among various roles and elements. These elements include industrial equipment, sensors, network connections, monitoring/attack, malicious attackers, attackers, control terminals, receivers, remote

controllers, puppets, attack zombies, and target hosts. Firstly, elements such as "industrial equipment," "sensors," "control terminals," "receivers," and "remote controllers" in Fig. 2 represent actual components of networked control systems. They carry out functions related to information gathering, data processing, and command execution. The graphic design of these elements is based on depictions created legitimately for actual devices and is in the public domain. Secondly, the "network connections" show the communication links between various components within the network control system. This design is a symbolic representation based on a simplified version of common network topology structures. "Monitoring/attack," "malicious attackers," "attackers," "puppets," and "attack zombies" depict the actors and techniques involved in DDoS attacks. "Monitoring/attack" may symbolize the attacker's monitoring and attacking actions against the target system. "Malicious attackers" and "attackers" refer to the unauthorized individuals initiating the attack. "Puppets" and "attack zombies" represent infected hosts controlled by the attacked party. They are typically incorporated into a botnet through malicious software to assist in executing DDoS attacks. Finally, the "target host" represents the ultimate goal of the DDoS attack—the server or network facility hosting the service that the attack aims to disrupt. Criminals call a lot of regular services from a lot of zombie hosts, which renders the service inoperable. A huge number of hosts are used in this type of attack, which is harder to defend against and has a greater success rate. When these types of network attacks happen, they are typically accompanied by a deluge of access requests, an increase in the access rate, the appearance of new Internet Protocol (IP) addresses, a decline in the number of original users, and sometimes even the appearance of data packets with special fields and functions. These anomalies represent the global state of the current network, which can help us judge whether a DoS attack is taking place or not and can also help people trace back to the IP address of the attacker.

## DDoS attack detection system based on deep learning

An attack detection system is required in order to quickly determine whether the network system is being attacked by DoS. One of those that is particularly challenging to identify is a DDoS attack. The network or system state will change when this assault happens, making it possible to identify DDoS attacks based on their unique characteristics. Among them, the detection method of machine learning, is to transform the detection problem into the problem of categorizing network data by using classifiers, and divide network traffic into attack traffic and normal traffic, but the detection effectiveness is low. Deep learning, on the other hand, automatically extracts sample features, which is more advantageous when the volume of data is considerable. CNN has a positive impact on image processing. Therefore, this article converts the original traffic into the data form equal to the image on the basis of the original traffic, further processes the data, then utilizes the deep learning model to train, and ultimately completes the detection of DDoS attacks. The three stages of data processing, model training, and attack detection make up the total attack detection architecture. The schematic is displayed in Fig. 3.

The data is handled in two directions at Fig. 3's data processing stage. The first step is to convert the original traffic data into binary values, with each data frame producing a

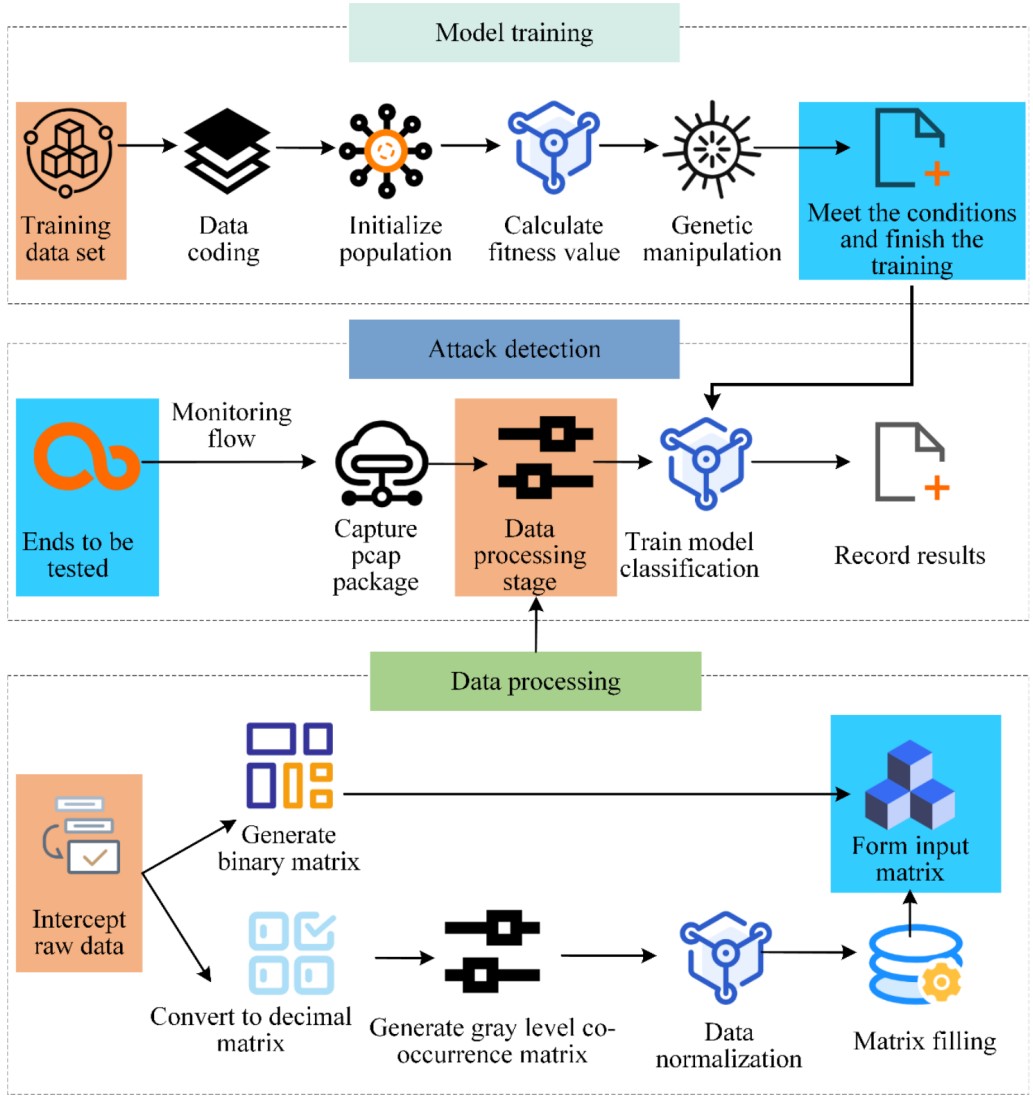

**Figure 3 Overall framework flow of attack detection method.** Data processing phase, processing data from two directions. The first step is to convert the raw traffic data into binary values, with each data frame generating the corresponding binary bit string before generating the binary matrix. Raw traffic data can also be used to generate decimal numbers based on bytes, with each data frame generating a string of decimal numbers that are combined to form a decimal matrix. In the stage of model training, the topology of CNN is designed by using the improved genetic algorithm, and the existing data sets are used for training. In the attack detection step, a trained neural network model is used to detect attacks and record classification results.

corresponding binary bit string before producing a binary matrix. The original traffic data can also be used to generate decimal numbers according to bytes, with each data frame producing a string of decimal numbers that come together to form a decimal matrix. The modified genetic algorithm is utilized to design the topology of the CNN during the model training stage, where the already-existing data sets are used for training. The trained

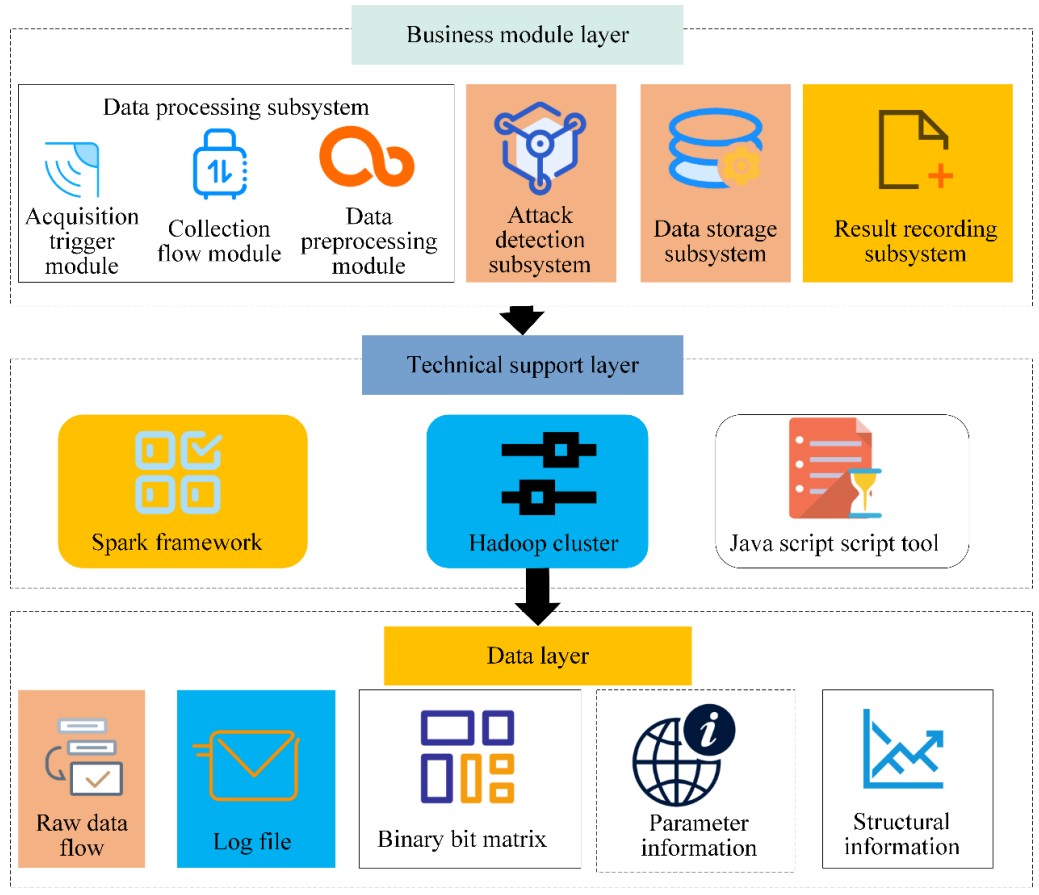

**Figure 4** **Attack detection system architecture.** The design of DDoS attack detection system includes the following features: capture and collect network traffic, pre-process the collected traffic, use genetic algorithm to determine the network structure parameters of CNN to train the model, and use the trained CNN to detect the attack traffic.

neural network model is used in the attack detection step to find attacks and record the classification outcomes.

This article proposes the design of a DDoS attack detection system based on deep learning, which can detect the DDoS traffic in the network to be detected in time and accurately, and then send an alarm signal to the system in time, so that the detection system can react in time and record the attack traffic in the log file. However, preventing DDoS attacks and detecting all traffic at once takes a lot of time. As a result, when the network system notices that the network traffic has suddenly changed significantly, it can conserve computational and storage resources by returning to the network attack detection state. As a result, the design of the DDoS attack detection system incorporates the following features: capturing and collecting network traffic, preprocessing the traffic collected, using genetic algorithm to determine the network structure parameters training model of CNN, and employing the trained CNN to detect attack traffic. Figure 4 displays the particular structure.

## Attack detection method of incomplete samples based on GAN

In the real network system environment, different attack methods will lead to the loss of network packets. These lost data packets will lead to abnormal and incomplete traffic data, which will lead to missing and abnormal feature attributes extracted from network traffic. These incomplete network traffic eigenvalues will directly affect the detection performance of existing network system attack detection models. The main structure of generative adversarial network (GAN) includes a G (Generator) and a D (Discriminator). The expression is:

$$\min_{G} \max_{D} V(D,G) = \mathbb{E}_{\boldsymbol{x} \sim p_{\text{data}}(\boldsymbol{x})}[\log D(\boldsymbol{x})] + \mathbb{E}_{\boldsymbol{z} \sim p_{z}(\boldsymbol{z})}[\log(1 - D(G(\boldsymbol{z})))] \tag{1}$$

where $D$ and $G$ represent Generator and Discriminator networks, which can be understood as the mapping of two functions. $D$ is used to judge whether the data is true or not, that is, the probability $D(x)$ that data X is true is obtained, and G generates data $G(z)$ from noise $Z$. $V$ is Value, which can be understood as Variation between real samples and generated samples. $\min_G \max_D$ means: firstly, $G$ is fixed, and adjust the parameters of $D$ so that it can judge whether the sample is real or generated to the maximum extent. Then, $D$ is fixed, and the parameters of $G$ are adjusted to minimize the difference between the generated sample and the real sample. By repeating $G$, data very close to the discriminator $D$ can be generated, and $D$ has better discriminating ability.

Wasserstein generative adversarial network-gradient penalty (WGAN-GP) is put forward to solve the existing problems of Wasserstein generative adversarial network (WGAN), and the Lipschitz constraint condition is constrained by gradient penalty instead of weight clipping method. Combining the advantages of Conditional GAN (CGAN) to specify the class samples, this article proposes an improved Conditional Wasserstein GAN-GP (CWGAN-GP), with the objective function as follows:

$$\mathcal{L} = \underbrace{\mathbb{E}_{\dot{\mathbf{x}} - \mathbb{P}_{\mathbf{z}}}[D(\dot{\mathbf{x}}|\mathbf{y})] - \mathbb{E}_{\mathbf{x} - \mathbb{P}_{r}}[D(\mathbf{x}|\mathbf{y})]}_{\text{Original critic loss}} + \underbrace{\lambda \mathbb{E}_{\dot{\mathbf{x}} - \mathbf{P}_{\mathbf{x}}}\left[\left(\left\|\nabla_{\dot{\mathbf{x}}} D(\dot{\mathbf{x}}|\mathbf{y})\right\|_{2} - 1\right)^{2}\right]}_{\text{Gradien penalty}} \tag{2}$$

$$\dot{\mathbf{x}} = G(\mathbf{z}|y) \tag{3}$$

$$\dot{\mathbf{x}} = \epsilon \mathbf{X} + (1 - \epsilon)\dot{\mathbf{x}} \tag{4}$$

where $\epsilon$ represents a random variable and $\epsilon \sim \mathbb{U}[0,1]$. $\lambda$ represents the penalty factor and is set to 10. $\mathbb{P}_{\dot{\mathbf{x}}}$ represents a uniform sampling distribution along a straight line between the true data distribution $\mathbb{P}_r$ and the false data distribution $\mathbb{P}_g$. Then, combining CWGAN-GP and WGANInverter, this article proposes an ICWGANInverter.

First, the training sample set $(X, Y)$ is used to train a CWGAN-GP, and then an inverse generator $\mathcal{I}_r$ is trained to inverse the given real sample $x$ to the potential space vector $z'$ of the pre-trained CWGAN-GP model, and then the generator $\mathcal{G}_\theta$ of CWGAN-GP can reconstruct the image $\dot{\mathbf{x}}$ from the inverse potential coding $z'$. The loss function of

CWGAN-GP is composed of discriminator loss, generator loss and reconstruction loss, and its discriminator loss function is:

$$\mathcal{L}_{\mathrm{dis}} = \underbrace{\mathop{\mathbb{E}}_{\mathbf{x}=\mathbb{P}_s}\big[\mathcal{D}_\varphi(\dot{\mathbf{x}}|\mathbf{y})\big] - \mathop{\mathbb{E}}_{\mathbf{x}-P_p}\big[\mathcal{D}_\varphi(\mathbf{x}|\mathbf{y})\big]}_{\text{Original critic loss}} + \lambda \underbrace{\mathop{\mathbb{E}}_{\dot{\mathbf{x}}-\mathbb{P}_{\dot{\mathbf{x}}}}\Big[\big(\big\|\nabla_{\dot{\mathbf{x}}}\mathcal{D}_\varphi(\acute{\mathbf{x}}|\mathbf{y})\big\|_2 - 1\big)^2\Big]}_{\text{Gradient penally}} \tag{5}$$

The loss function $\mathcal{L}_{gen}$ of the generator is calculated as follows:

$$\mathcal{L}_{gen} = -\mathop{\mathbb{E}}_{z \sim p_t(z)}\big[\mathcal{D}_\varphi\big(\mathcal{G}_\theta(z|y)\big)\big] \tag{6}$$

The reconstruction loss $\mathcal{L}_{\mathrm{rec}}$ is calculated as follows:

$$\mathcal{L}_{\mathrm{rec}} = \min_y \mathbb{E}_{x-p_x(x)}\big\|\mathcal{G}_\theta\big(\mathcal{I}_y(x)|y\big) - x\big\| + \lambda \cdot \mathbb{E}_{z-p_z(=)}\big[\mathcal{L}\big(z, \mathcal{I}_\gamma\big(\mathcal{G}_\theta(z|y)\big)\big)\big] \tag{7}$$

$$\dot{\mathbf{x}} = \mathcal{G}_\theta(z|y) \tag{8}$$

where $\mathbb{P}_r$ represents the distribution of real sample data, $\mathbb{P}_g$ represents the distribution of generated sample data, and $\mathbb{P}_{\dot{\mathbf{x}}}$ represents the uniform sampling from the straight line between two points of real data distribution $\mathbb{P}_r$ and generator generated data distribution $\mathbb{P}_g$. If $L_2$ distance is used as the loss of $\mathcal{L}\big(z, \mathcal{I}_\gamma\big(\mathcal{G}_\theta(z|y)\big)\big)$, the $\lambda$ value is 0.1. If Jensen–Shannon (JS) divergence is used as the loss of $\mathcal{L}\big(z, \mathcal{I}_\gamma\big(\mathcal{G}_\theta(z|y)\big)\big)$, $\lambda$ is 1.

Then ICWGANInverter is used to build a DoS attack detection model, which automatically learns the high-level abstract information of the original data, and then uses the reconstruction error method to identify the best classification label. The specific architecture diagram is shown in Fig. 5.

In Fig. 5, the original test sample and the prediction category label are utilized as the input of the model, and the input test sample can be reconstructed after the prediction category label of the test sample with partial characteristics is achieved in feature reconstruction. In specifically, the attacker wants to lower the far-end estimator's estimation accuracy by causing packet loss through the assault. DoS attacks can involve obstructing communication lines or sending a lot of pointless data packets. An attacker who can listen in on the system channel can find out some important information that is transmitted there by watching the channel and using other methods.

## Experimental design

The NSL-KDD dataset is utilized as intrusion detection data to test the efficacy of the aforementioned DoS attack detection approach. Despite the fact that this data set might not accurately reflect the current real network, it can nevertheless be used as a benchmark to help researchers compare various intrusion detection strategies. Also, there are no duplicate records in either the training set or the test set, and the distribution of data across both sets is more evenly distributed. To assess the efficacy of the aforementioned model and contrast it with other well-known network attack detection models, some discrete properties of the dataset NSL-KDD are eliminated from the experiment. Protocol, flag, and service are the three symbolic elements included in the NSL-KDD dataset. By removing one

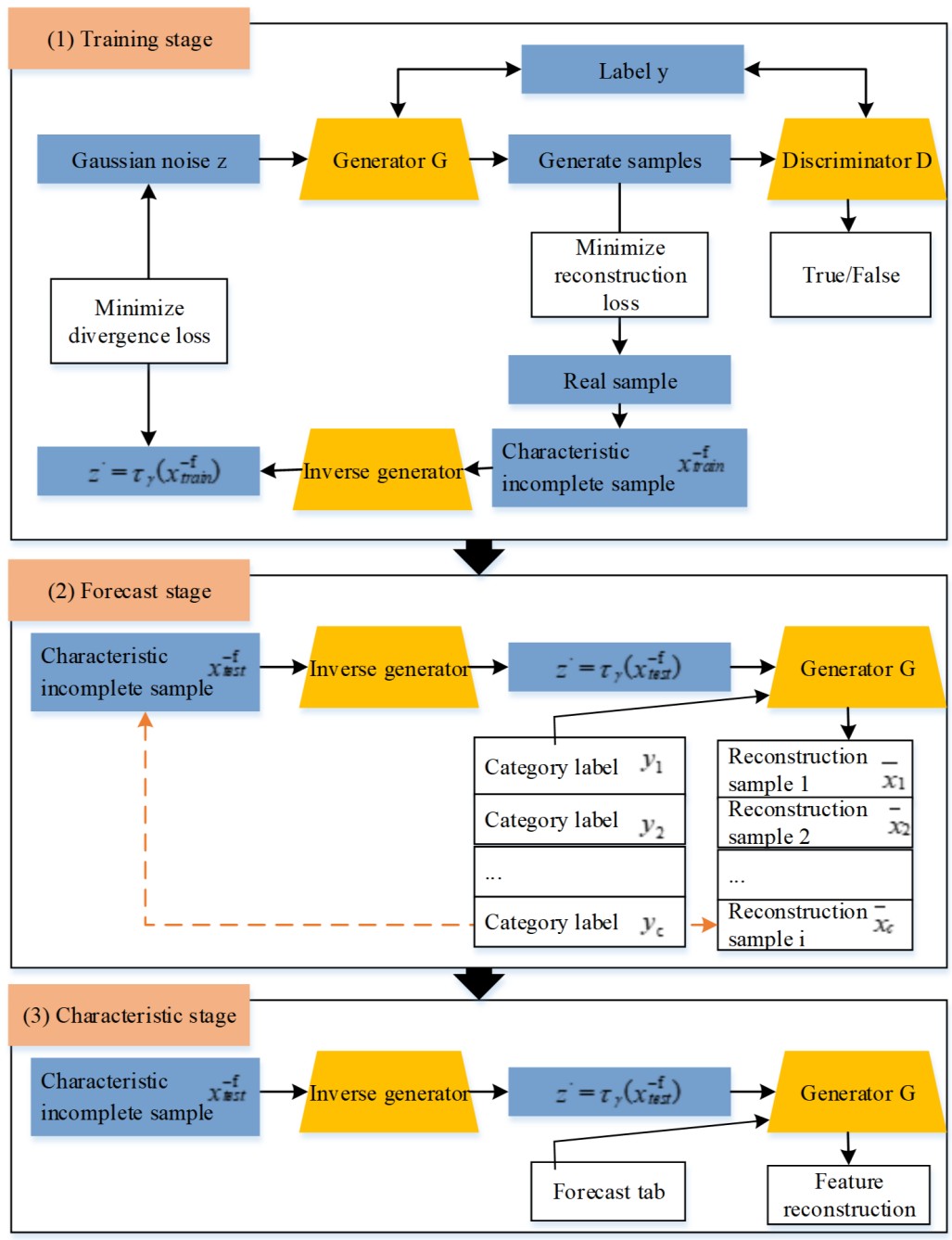

**Figure 5  ICWGANInverter incomplete feature DoS attack detection architecture diagram.**

of the feature values at random, an incomplete feature data set is created. KDDTest+ and KDDTest-21 are two of the NSL-KDD dataset's sub-datasets. The NSL-KDD dataset offers several advantages over the original KDD dataset: it does not contain redundant records in the training set, preventing classifiers from favoring more frequent records. The proposed test set does not include duplicate records, ensuring that the learner's performance is not

biased towards methods with better detection rates for frequent records. The number of records selected from each difficulty level group is inversely proportional to the percentage of records in the original KDD dataset. This results in a larger variation in classification rates for different machine learning methods, making it more effective to accurately assess different learning techniques. The number of records in both the training and test sets is reasonable, allowing experiments to be run on the entire set without the need for random selection of a small portion. Consequently, evaluation results from different research efforts are consistent and comparable. The NSL-KDD dataset includes the following files:

KDDTrain+.ARFF: The complete NSL-KDD training set with binary labels in Attribute-Relation File Format (ARFF) format.

KDDTrain+.TXT: The complete NSL-KDD training set, including Comma Separate Values (CSV) formatted attack type labels and difficulty levels.

KDDTrain+_20Percent.ARFF: A 20% subset of the KDDTrain+.arff file.

KDDTrain+_20Percent.TXT: A 20% subset of the KDDTrain+.txt file.

KDDTest+.ARFF: The complete NSL-KDD test set with binary labels in ARFF format.

KDDTest+.TXT: The complete NSL-KDD test set, including CSV formatted attack type labels and difficulty levels.

KDDTest-21. ARFF: A subset of the KDDTest+.arff file excluding records with difficulty level 21 (out of 21 levels).

KDDTest-21.TXT: A subset of the KDDTest+.txt file excluding records with difficulty level 21 (out of 21 levels).

The dataset can be accessed at: https://www.kaggle.com/datasets/hassan06/nslkdd. The NSL-KDD dataset includes five types of network data: Normal, DoS, User to Root (U2R), Remote to Local (R2L) and Probe. The data can be found at Mendeley: DOI 10.17632/pcfccfnvj2.1.

In this article, KDDTest+ and KDDTest-21 datasets have the problem of category imbalance, that is, the number of positive (DoS attack) and negative (non-attack) samples is quite different. In order to deal with this challenge effectively, the method of adjusting the category weight is adopted, and the sample weight is adjusted when training the deep learning model, so that the model pays more attention to a few categories, thus improving the accuracy of its recognition. Specifically, the concept of class weights is used to balance the attention of the model to different types of samples in the training process by giving different weights to different categories. This method can introduce the weight term into the loss function, which makes the model pay more attention to the learning of a few categories, thus improving the performance on unbalanced datasets. The calculation of category weight is usually based on the frequency of category, and the specific calculation is shown in (Eq. 9):

$$w_c = \frac{N}{K \times n_c} \tag{9}$$

In Eq. 9, $w_c$ is the weight of category $c$. $N$ is the total number of samples, $K$ is the total number of categories, and $n_c$ is the number of samples of category $c$. In the training process, these category weights are used for the loss items of the corresponding categories,

so that the model can learn the characteristics of each category more evenly. This strategy of weight adjustment is helpful to improve the sensitivity of the model to a few categories to better adapt to the imbalance of data. In the experiment of this article, the category weights are calculated according to KDDTest+ and KDDTest-21 datasets, and applied to the training process of deep learning model to improve the detection performance of DoS attacks.

The evaluation indexes are accuracy, recall, precision, false positive rate (FPR), F1-score, G-mean and Area Under ROC Curve (AUC).

The calculation equation is as follows:

$$\text{FPR} = \frac{FP}{TN + FP} \tag{10}$$

$$\text{Accuracy} = \frac{TP}{TP + FP} \tag{11}$$

$$\text{Recall} = \frac{TP}{TN + FP} \tag{12}$$

$$\text{Precision} = \frac{FP}{TN + FP} \tag{13}$$

$$\text{F1} = \frac{2 * \text{Precision*Recall}}{\text{Precision} + \text{Recal}}. \tag{14}$$

In order to evaluate the performance of ICWGANInverter model more comprehensively, Matthews correlation coefficient (MCC) is introduced as an additional evaluation index. MCC is an evaluation standard that comprehensively considers the indexes of binary confusion matrix, and its range is between −1 and 1. The closer the value is to 1, the better the classification performance, 0 means random classification, and −1 means completely opposite classification. Equation 15 shows the calculation method of MCC:

$$\text{MCC} = \frac{TP \times TN - FP \times FN}{\sqrt{(TP + FP)(TP + FN)(TN + FP)(TN + FN)}}. \tag{15}$$

In the above equation, $TP$ is the number of positive samples correctly classified, $TN$ is the number of negative samples correctly classified, $FP$ is the number of positive samples wrongly classified, and $FN$ is the number of positive samples wrongly classified as negative samples.

## Experimental environment and parameter setting

The experiment runs in Windows 10 64-bit operating system (64 GB RAM, Intel 5-2620 CPU, ThinkStation workstation). Using Python programming language and TensorFlow deep learning framework, it is used to build, train and evaluate deep learning models.

**Table 1  MCC evaluation results of ICWGANInverter model.**

| Dataset | Model | TP | TN | FP | FN | MCC |
|---|---|---|---|---|---|---|
| | ICWGANInverter | 400 | 800 | 50 | 30 | 0.750 |
| KDDTest+ | *Wang et al. (2022)* | 350 | 780 | 70 | 40 | 0.680 |
| | *Aldhyani & Alkahtani (2023)* | 380 | 790 | 60 | 35 | 0.720 |
| | ICWGANInverter | 300 | 700 | 100 | 50 | 0.650 |
| KDDTest-21 | *Wang et al. (2022)* | 280 | 720 | 80 | 60 | 0.610 |
| | *Aldhyani & Alkahtani (2023)* | 320 | 710 | 90 | 55 | 0.630 |

Pandas and Matplotlib are used to process and visualize experimental data. Jupyter Notebooks are used to organize and run the experimental code in an interactive way, which is convenient for the visualization and debugging of the experimental process. $(n_x, n_z \times 8, n_z \times 4, n_z \times 2, n_z)$; The number of units in each layer of the decoder $(n_z + n_y, n_z \times 2, n_z \times 4, n_z \times 8, n_x)$; Number of units in each layer of discriminator (40,20,10). In order to ensure the repeatability of the experiment and the reliability of the results, the following hyperparameters in Table 1 are used when training the ICWGANInverter model:

## RESULTS AND DISCUSSION

### Detection results of continuous incomplete feature attacks

On the NLS-KDD dataset, the continuous eigenvalues are noisy, and the noisy training set and test set are created. The ICWGANInverter model is trained using the noisy data set after performing feature reconstruction and classification under various noise factors. The mean square error is obtained when the eigenvalues of the reconstructed continuous features are tested on the sub-datasets KDDTest+ and KDDTest-21, and the results are displayed in Fig. 6.

In Fig. 6, with the increase of noise factor, the mean square error of continuous feature reconstruction of ICWGANInverter model on KDDTest+ and KDDTest-21 noisy test sets also increases gradually, which shows that the feature reconstruction ability of ICWGANInverter model is inversely proportional to the noise intensity. When λ value is 0, there is no noise and the model error is the smallest. When λ value is 1, there is no noise and the model error is the largest. It shows that the model has approximately equal feature reconstruction ability for different noises and has high robustness. The overall test performance is shown in Fig. 7.

In Fig. 7, with the increase of the value of noise factor λ, the values of Accuracy, Recall, Precision, F1-score, G-mean and FPR all fluctuate to a certain extent, and generally show a downward trend. It shows that the stronger the noise, the lower the attack detection performance of the test samples, which shows that ICWGANInverter model is sensitive to the deviation of continuous eigenvalues.

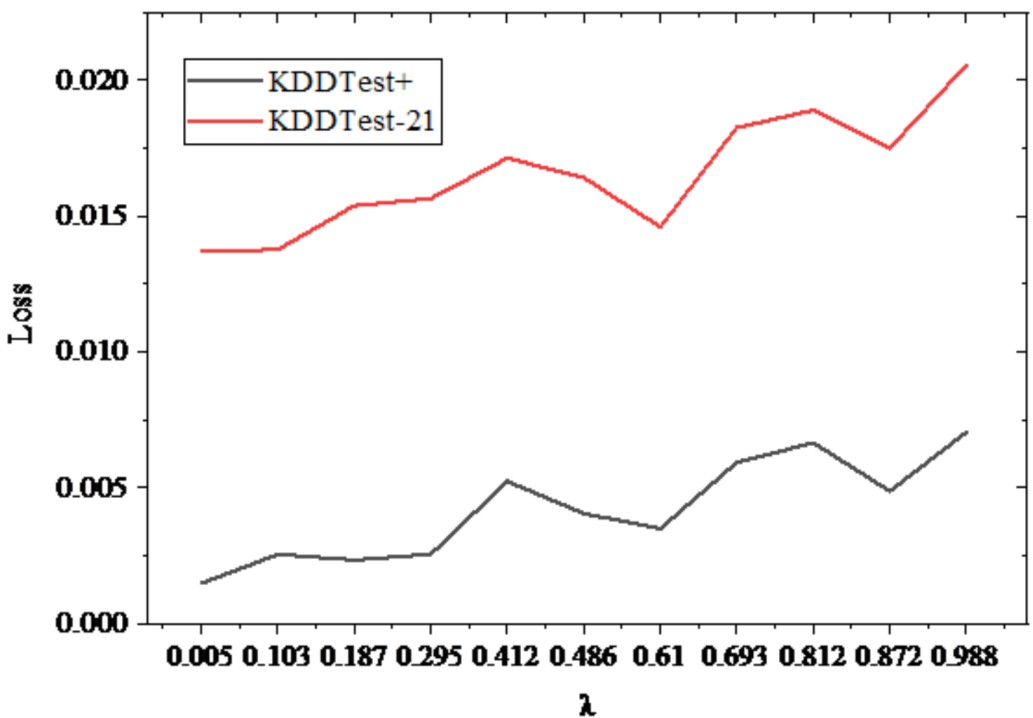

**Figure 6** Mean square error loss of continuous feature reconstruction under different noise factors $\lambda$ .

On KDDTest+ and KDDTest-21 noisy test sets of ICWGANInverter model, the receiver operating characteristic curve (ROC) and AUC value under different noise factors are shown in Figs. 8 and 9.

The overall AUC values of macro-average and micro-average are over 0.8 in Figs. 8 and 9, and the ROC curves are shown in the upper end of the diagonal. This shows that the ICWGANInverter model has good detection performance in both single-class attack detection and overall attack detection. Moreover, the AUC value of a single category decreases with increasing noise factor values, demonstrating that detection performance declines with increasing noise intensity.

## Performance comparison of different attack detection models

In order to further verify the DoS attack detection performance of the ICWGANInverter model proposed in this article, the performance of similar methods proposed in *Wang et al. (2022)* and *Aldhyani & Alkahtani (2023)* on NSL-KDD noisy ($\lambda = 0$) datasets is compared. The result is shown in Fig. 10.

Figure 10's results demonstrate that, in comparison to other models, the ICWGANInverter model put forward in this article has an accuracy of 87.79%, a recall value of 83.79%, an F1-score value of 87.37%, and an FPR value of 8.26% on KDDTest+ data sets. For the KDDTest-21 dataset, the DoS attack detection accuracy is 75.93%, the recall value is 77.15%, the F1-score value is 83.10%, and the FPR value is 28.16%. The advantages

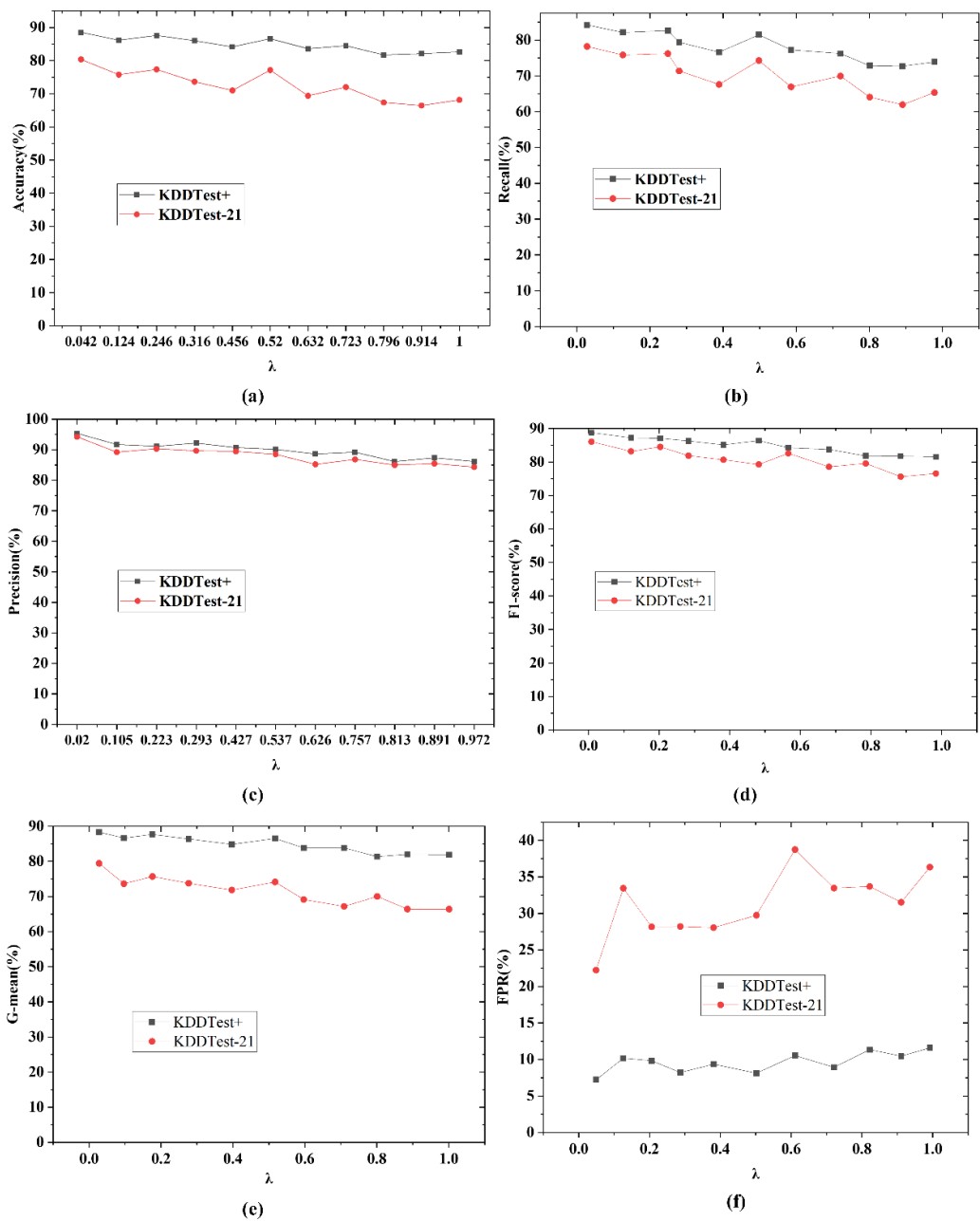

**Figure 7** **The overall performance under different noise factors λ.** (A) Accuracy; (B) recall; (C) precision; (D) F1-score; (E) G-mean; (F) FPR.

of the detection model in this article are demonstrated by the fact that it outperforms the detection models in *Wang et al. (2022)* and *Aldhyani & Alkahtani (2023)* in every way.

MCC evaluation of ICWGANInverter model shows that the model has excellent DoS attack detection performance on KDDTest+ and KDDTest-21 data sets. Compared with the models in *Wang et al. (2022)* and *Aldhyani & Alkahtani (2023)*, the proposed model shows obvious advantages in many evaluation indexes. In terms of Matthews correlation

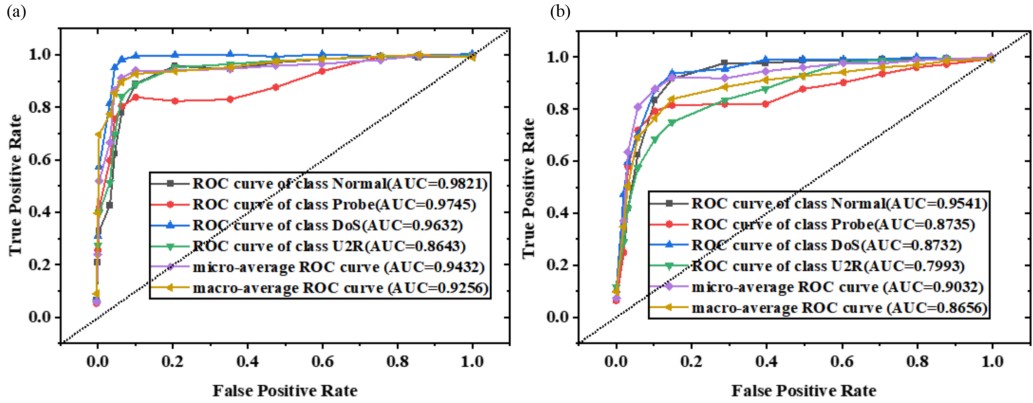

**Figure 8** ROC curve and AUC value of the model on KDDTest+ noisy test set ((A) λ =0; (B) λ =0.5).

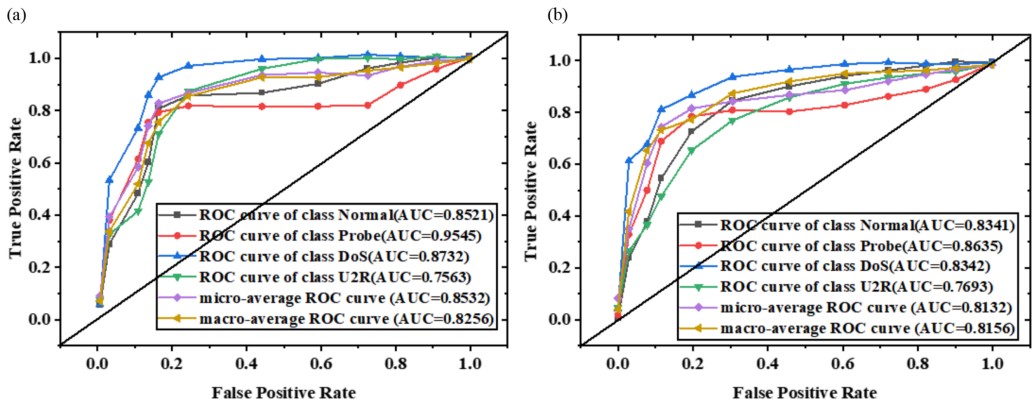

**Figure 9** ROC curve and AUC value of the model on KDDTest-21 noisy test set ((A) λ =0; (B) λ =0.5).

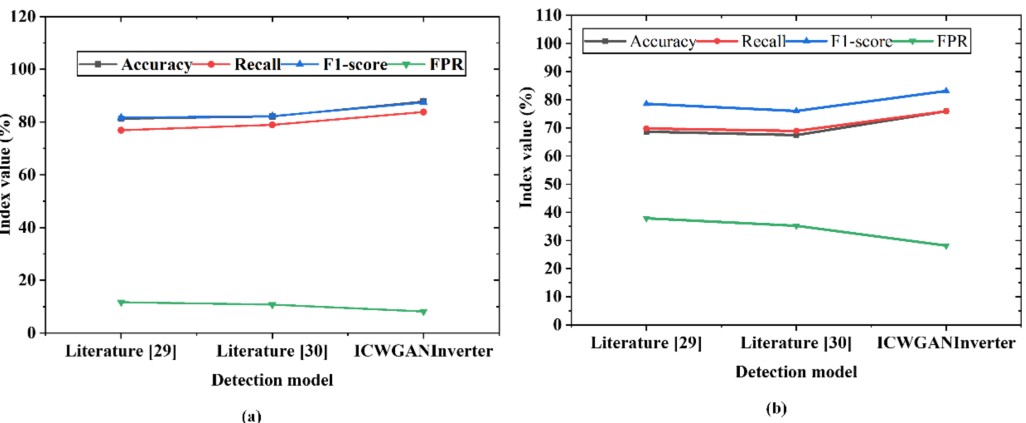

**Figure 10** Performance comparison of different detection models on noisy test set ((A) KDDTest+; (B) KDDTest-21).

coefficient (MCC), the MCC value of ICWGANInverter model on KDDTest+ data set is 0.750, which is significantly higher than 0.680 in *Wang et al. (2022)* and 0.720 in *Aldhyani & Alkahtani (2023)*. On the KDDTest-21 dataset, the MCC value of this model is 0.650, which is also higher than the other two models (0.610 and 0.630 respectively). These results fully prove the excellent performance of ICWGANInverter model in dealing with distributed DoS attack traffic.

In addition, the training time of the model on KDDTest+ dataset is 12 min, the detection time is 5.6 ms, and the detection speed reaches 180 samples per millisecond. On KDDTest-21 dataset, although the training time is slightly longer (14 min) and the detection time is increased to 8.2 ms, the detection speed is still as high as 150 samples per millisecond. This is a very rare and valuable performance in the current DoS attack detection field. To sum up, ICWGANInverter model is superior to the existing similar research in comprehensive performance of DoS attack detection, which not only shows outstanding detection accuracy, but also has advantages in practical application efficiency.

Figure 11 shows the detection results of different types of data in NSL-KDD dataset by different models. ICWGANInverter model performs well in Normal type, reaching the detection accuracy of 97.21%, but it drops slightly in DoS type, reaching 84.77%. On the types of Probe and R2L, the performance of ICWGANInverter model is obviously lower than the other two algorithms, which are 66.25% and 21.37% respectively. On U2R type, the detection accuracy of ICWGANInverter model is 7.00%. In contrast, *Wang et al. (2022)* performs relatively well in Normal and Probe types, accounting for 97.37% and 68.86% respectively, but its performance in DoS and R2L types is poor, accounting for 81.47% and 2.73% respectively. *Aldhyani & Alkahtani (2023)* has a relatively high accuracy rate (97.27%) in Normal type, but the detection accuracy rate in DoS, Probe and U2R types is relatively low, which are 79.38%, 62.21% and 4.50% respectively. Different algorithms have obvious differences in different types, which shows that various classification algorithms do not have advantages for all types of data, but each has its own advantages. Considering the advantages of each algorithm, people can try to optimize the combination of algorithms in the future to improve the overall detection effect.

The detection performance of ICWGANInverter model on NSL-KDD dataset for different types of network attacks and normal traffic is evaluated separately. In particular, the following five types of data are analyzed in detail: Normal, DoS, U2R, R2L and Probe. For each type of network data, a confusion matrix is generated to visually show the performance of the model.

There are 1,250 samples actually classified as DoS attacks, 1,200 samples correctly predicted as DoS attacks by ICWGANInverter model, and 50 samples predicted as non-DoS attacks. There are 1,250 samples which are actually classified as non-DoS attacks, 1,180 samples are correctly predicted as non-DoS attacks and 70 samples are predicted as DoS attacks by ICWGANInverter model. Similarly, a confusion matrix will be constructed for normal traffic, U2R, R2L and Probe attacks. Table 2 shows the confusion matrix under different conditions:

According to the confusion matrix, the calculated evaluation indexes include accuracy, recall, precision, F1 score, false positive rate (FPR) and G-means. The following are the

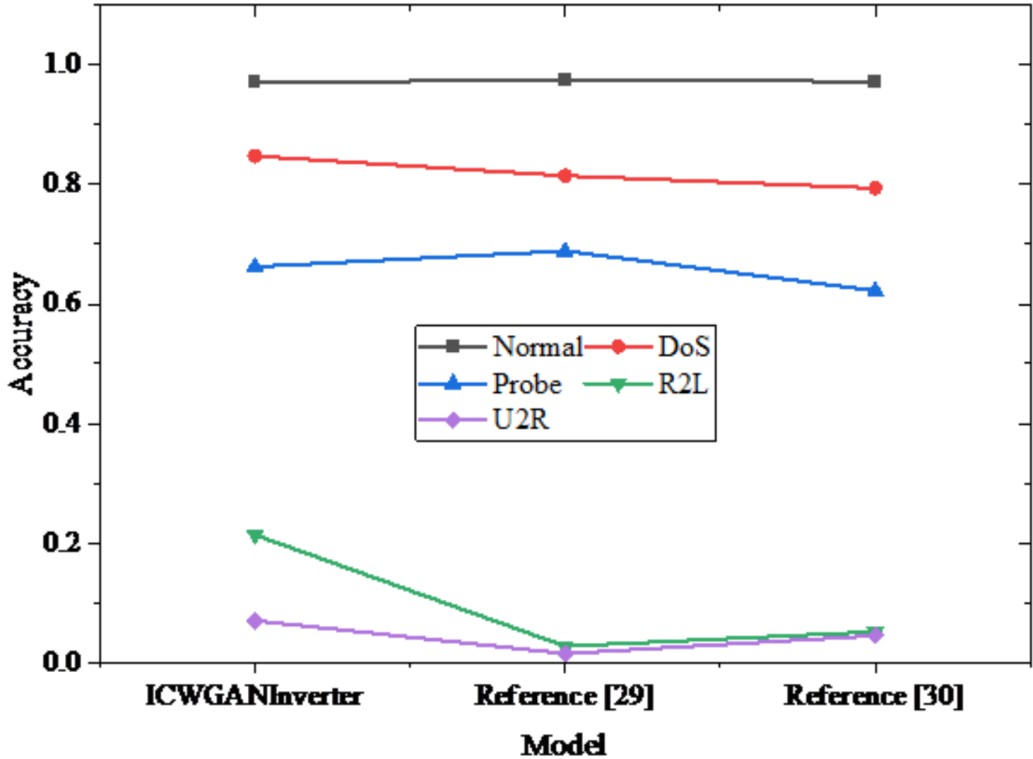

**Figure 11** Detection results of different models in different types of data in NSL-KDD dataset.

**Table 2** The detection speed and running time of this model on KDDTest+ and KDDTest-21 subdatasets.

| Dataset | Model | Training time (min) | Detection time (ms) | Detection speed (/ms) |
|---------|-------|---------------------|---------------------|-----------------------|
| KDDTest+ | ICWGANInverter | 12 | 5.6 | 180 |
| KDDTest-21 | ICWGANInverter | 14 | 8.2 | 150 |

evaluation results of various traffic and attacks: normal traffic: accuracy: 96.29%, recall: 96.43%, precision: 95.00%, F1 score: 95.21%, false alarm rate: 5.38%, G-means: 95.77%. U2R attack: accuracy: 97.86%, recall: 84.62%, precision: 78.57%, F1 score: 81.75%, FPR: 2.36%, G-means: 92.07%. R2L attack: accuracy: 98.57%, recall: 90.00%, precision: 81.82%, F1 score: 85.33%, FPR: 1.59%, G-means: 94.52%. Probe attack: accuracy: 97.50%, recall: 79.31%, precision: 85.19%, F1 score: 82.86%, FPR: 1.60%, G-mean: 89.07. To sum up, ICWGANInverter model is excellent in detecting different types of network attacks and normal traffic. Specifically, the model performs best in detecting normal traffic and R2L attacks, followed by U2R attacks and Probe attacks.

## DISCUSSION

The research mentioned above demonstrates that the detection model put forward in this article performs better than other models of a similar nature. Convergence speed

and detection accuracy both exhibit a positive trend, indicating that GAN can be relied upon to identify DoS attacks. However, further optimization is required to improve the effectiveness of the detection process, in line with the findings of *Velliangiri, Karthikeyan & Kumar (2021)*. This article gathers customer service requests and classifies them into log data to detect DDoS assaults using a classifier built using deep learning. The greatest detection accuracy is 0.830, and Bhattacharya distance measurement is then used to choose a few key features from the log file to shorten the classifier's training period. A new architecture with two components—DoS/DDoS detection and mitigation—was proposed by *Mihoub et al. (2022)*. Due to its ability to recognize particular attack types and the packet types that were utilized in them, the detection component offers fine-grained detection. This makes it possible to use corresponding mitigation strategies to put countermeasures in place for specific packet types. A DNN approach for real-time detection of DDoS assaults in software-defined networks was put forth by *Makuvaza, Jat & Gamundani (2021)*. The suggested strategy generates 97.59% detection accuracy while using less time and resources. It demonstrates that the detection accuracy of various comparable detection methods varies, and the benefits of this article are also shown through comparison. On the one hand, it is important to constantly scan the current network master nodes, look for any potential security holes, and patch any newly discovered vulnerabilities as soon as they are discovered to effectively avoid DoS attacks. On the other hand, if an attack is discovered, it may be focused towards a few sacrificial hosts to shield the true host from harm.

This study focuses on the field of intrusion detection and draws lessons from the research results of relevant scholars. *Luo, Huang & Li (2023)* used various input feature subsets of KDDTest+ and KDDTest-21 datasets as the input features of support vector machine (SVM) classifier (*Luo, Huang & Li, 2023*). According to the experimental results, the proposed technology achieved 88.25% classification accuracy when only using KDDTest+ dataset, and 72.42% classification accuracy when using KDDTest-21. This research is related to the research of this article, because it discusses the technology and methods in the field of intrusion detection, especially the method of feature selection and classification merging using SVM. In the research, the model based on GAN is mentioned, while the scholar focused on SVM. *Andalib & Vakili (2020)* used three learning technologies at the same time: gated cycle unit, CNN as depth technology and random forest as integration technology. These systems were trained in parallel on NSL-KDD dataset (*Andalib & Vakili, 2020*). The simulation results showed that the accuracy was 87.28% on NSL-KDD "KDDTest+" dataset, "KDDTest-21". This is related to the research of this article, because they all discuss the methods to improve the intelligence and performance of intrusion detection system, especially paying attention to the autonomy in the face of zero-day attacks. The innovation of this article lies in the introduction of ICWGANInverter model. Through experiments on NLS-KDD datasets, the feature reconstruction and classification performance of this model are tested under different noise factors. The chart shows the mean square error of continuous feature reconstruction of ICWGANInverter model on KDDTest+ and KDDTest-21 noise test sets when the noise factor increases gradually. The results show that the feature reconstruction ability of the model is inversely proportional to the noise intensity and has high robustness to different noises.

In addition, in terms of overall performance, the ICWGANInverter model performs well in single-category attack detection and overall attack detection through the display of ROC curve and AUC value. Compared with the models proposed in references (*Wang et al., 2022*) and (*Aldhyani & Alkahtani, 2023*), the accuracy of 87.79%, Recall value of 83.79%, F1-score value of 87.37% and FPR value of 8.26% are achieved on KDDTest+ dataset. On the KDDTest-21 dataset, although the accuracy and recall decreased slightly, they still reached 75.93% and 77.15% respectively. This highlights that the ICWGANInverter model is superior to the reference model in all aspects, and proves the remarkable performance advantage of the method proposed in this article in detecting DoS attacks.

In addition, *Premkumar, Sundararajan & Mohanbabu (2022)* proposed a framework of hybrid intrusion detection system based on multi-layer perceptron strategy. The hybrid intrusion detection system used a variety of machine learning algorithms, including naive Bayes, random forest, decision tree, multi-layer perceptron, K nearest neighbor and support vector machine, to improve the detection effect (*Premkumar, Sundararajan & Mohanbabu, 2022*). The experimental accuracy of NSL-KDD dataset and UNSW-NB15 dataset was 99.963% and 98.751%, respectively, but this method was not a confirmation method of various DoS attacks. Awad et al. (2023) used hybrid particle swarm optimization algorithm and chaotic butterfly optimization algorithm to optimize the weight of LSTM algorithm, and used the improved version of LSTM algorithm to test on NSL-KDD and LITNET-2020 datasets (*Awad, Ali & Gaber, 2023*). The results showed that the accuracy reached 93.09% and the accuracy was 96.86%. However, this article focuses on the ICWGANInverter model. Through experiments on NLS-KDD datasets, the model has high robustness to different noises, but the detection performance drops when it faces large noises. On KDDTest+ and KDDTest-21 noise test sets, the model performs well, and the AUC value exceeds 0.8, which shows good performance in single-class attack detection and overall attack detection.

Comparing the performance of different attack detection models, compared with the models in references (*Wang et al., 2022*) and (*Aldhyani & Alkahtani, 2023*), the ICWGANInverter model proposed in this article has achieved 87.79% accuracy, 83.79% recall, 87.37% F1-score and 8.26% FPR on KDDTest+ datasets. For KDDTest-21 dataset, the accuracy of DoS attack detection is 75.93%, the recall rate is 77.15%, the F1-score is 83.10%, and the FPR value is 28.16%. This highlights that the method proposed in this article is superior to the reference model in all aspects. Generally speaking, this article deeply discusses the performance and innovation of intrusion detection system by introducing ICWGANInverter model, and pays special attention to the model robustness in the face of zero-day attacks.

## CONCLUSION

DoS attacks overtax the system's critical resources and keep it too busy to provide useful services. Typically, intrusion detection, traffic filtering, and multiple authentications are used as DoS attack defense strategies. Filtered traffic that aims to reduce network bandwidth will be allowed to proceed normally. As a result, this article initially examines the fundamentals and attack tactics of DoS attacks before looking into the most recent

DoS attack detection techniques. A distributed DoS attack detection system based on deep learning is established in response to the investigation's limitations. This system can quickly and accurately identify the traffic of distributed DoS attacks in the network that needs to be detected and then promptly send an alarm signal to the system. Then, in order to address the insufficient network traffic characteristics in DoS attacks, an improved model called the ICWGANInverter model based on GAN is put forward. This model automatically learns the advanced abstract information of the original data and then employs the method of reconstruction error to identify the best classification label. It is then tested on the intrusion detection dataset NSL-KDD. The outcomes demonstrate the excellent accuracy of the suggested strategy in identifying DoS attacks. In addition, it is found that ICWGANInverter model shows high accuracy and low false positive rate in DoS attack detection, which shows that this model has good recognition ability for DoS attacks. For U2R and R2L attacks, although the model has also achieved good performance, the recall rate of the model is slightly lower because of the relatively small number of samples of these two types of attacks. The detection results of Probe attacks and normal traffic also prove the effectiveness and stability of the model. The article does, however, have several shortcomings. For instance, only a small collection of data is used to verify the strategy suggested in this article. Other datasets that are modeled following network penetration are not used. Future study will therefore concentrate on finding more situations to investigate the shortcomings and room for advancement of this technology to make it suitable for a range of network attacks.

### Funding

This work was supported by the Basic Scientific Research Projects of Central Universities, Research on network attack oriented forensics technology (No. 2022TJJBKY027), Research on recognition technology of refitted vehicles based on Artificial Intelligence (No. 2021TJJBKY023) and the Key scientific research projects of colleges and universities in Henan Province, Research and application of key technologies of open source information Mining (No. 23B520019). The funders had no role in study design, data collection and analysis, decision to publish, or preparation of the manuscript.

### Grant Disclosures

The following grant information was disclosed by the authors:
The Basic scientific research projects of central universities.
Research on network attack oriented forensics technology: No. 2022TJJBKY027.
Research on recognition technology of refitted vehicles based on Artificial Intelligence: No. 2021TJJBKY023.
The Key scientific research projects of colleges and universities in Henan Province.
Research and application of key technologies of open source information mining:  No. 23B520019.

## Competing Interests

The authors declare there are no competing interests.

## Author Contributions

- Yang Li conceived and designed the experiments, performed the experiments, performed the computation work, prepared figures and/or tables, authored or reviewed drafts of the article, and approved the final draft.
- Haiyan Wu conceived and designed the experiments, analyzed the data, performed the computation work, prepared figures and/or tables, and approved the final draft.

## Data Availability

The NSL-KDD dataset is available at Kaggle: https://www.kaggle.com/datasets/hassan06/nslkdd.

The CICEV2023 DDoS attack datasets are available at: https://www.unb.ca/cic/datasets/cicev2023.html.

## Supplemental Information

Supplemental information for this article can be found online at http://dx.doi.org/10.7717/peerj-cs.2162#supplemental-information.

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
