# Peer review of "A DoS attack detection method based on adversarial neural network"

_PeerJ Computer Science, doi:10.7717/peerj-cs.2162_

## Round 0.1 · original submission · Major Revisions

Based on three reviewers' comments, the paper needs a major overhaul. Most of the concerns raised by reviewers (particularly R2) are valid and should be incorporated before acceptance.

·

Basic reporting

This paper ûrst explores the principles and attack strategies of denial-of-service (DoS) assaults before looking into the current detection techniques for DoS attacks in order to investigate the impact of deep learning model on detecting DoS attacks. A distributed DoS attack detection system based on deep learning is established in response to the investigation's limitations. This work is meaningful.

Experimental design

1 It was suggested that author should show a flowchart of this work.
2 It was suggested that author should employ MCC in this work.
3 Some efforts can be discussed in this work.

Validity of the findings

A distributed DoS attack detection system based on deep learning is established in response to the investigation's limitations. This system can quickly and accurately identify the traffic of distributed DoS attacks in the network that needs to be detected and then promptly send an alarm signal to the system. Then, a model called the Improved Conditional Wasserstein Generative Adversarial Network with Inverter (ICWGANInverter) is proposed in response to the characteristics of incomplete network traffic in DoS attacks.

Additional comments

The language should be polished by fluent English speakers.

Reviewer 2 ·

Basic reporting

After revising the paper, I believe it requires major revisions for the following reasons:

1. The KDDTest+ and KDDTest-21 datasets are high degree of class imbalance. However, the author's paper did not address the approach taken to mitigate this imbalance within the dataset.

2. Author has mentioned that their system can quickly and accurately identify the traffic of distributed DoS attacks but without showing the running time and the detection time of the proposed model.

3. Some existing research papers has achieved slightly higher accuracy than author on KDDTest+ and KDDTest-21 respectively such as:

[1] L. Tianyao, H. Huadong and L. Run, "An Intrusion Detection Framework with Optimized Feature Selection and Classification Combination Using Support Vector Machine," 2023 IEEE 3rd International Conference on Electronic Technology, Communication and Information (ICETCI), Changchun, China, 2023, pp. 182-186, doi: 10.1109/ICETCI57876.2023.10176950.

[2] Andalib and V. T. Vakili, "An Autonomous Intrusion Detection System Using an Ensemble of Advanced Learners," 2020 28th Iranian Conference on Electrical Engineering (ICEE), Tabriz, Iran, 2020, pp. 1-5, doi: 10.1109/ICEE50131.2020.9260808.

While the discrepancy in accuracy between the author's work and other research is relatively small, it suggests that the existing methods are performing adequately, and the author's contribution may not introduce a significantly novel approach.

4. Moreover, it's worth noting that the research referenced in [1] and [2] has demonstrated successful detection of a broader range of cyberattacks beyond just Denial of Service (DoS) attacks.

5. Many prior research papers have used KDDTest+ and KDDTest-21 datasets. Therefore, it is important to conduct a thorough and equitable comparison between the author's research work and the most recent existing studies. This comparative analysis should demonstrate that the proposed model surpasses other established models in terms of performance and effectiveness.

6. The novelty of the paper is not clear.

7. The hardware and software setups used in our experiment are not mentioned.

8. There are other cyberattacks that is presented in the datasets such as U2R, R2L, and PROBE that are not mentioned or detected.

Experimental design

no comment

Validity of the findings

The novelty of the paper is not clear and some existing research papers has achieved slightly higher accuracy than author on KDDTest+ and KDDTest-21.

Cite this review as

Reviewer 3 ·

Basic reporting

1. The architecture and hyperparameters of the deep learning network is not clear (e.g. the number of layers, input/output dimension, optimizer), which may raise concerns regarding reproducibility.
2. Merge Figure 1 and 2, and add more traffic descriptions to Figure 2.
3. The related works lack of descriptions for GNN-based studies.

Experimental design

1. The manuscript mentions that the proposed model has the "quickly" feature, but there is no explanation of detection time delay in the experimental part.

Validity of the findings

1. The provided code lacks the implementation of proposed ICWGANInverter and CNN.

Cite this review as

---

## Round 0.2 · Major Revisions

Reviewer 4 has raised many fundamentally relevant questions. I suggest authors to incorporate the suggestions and resubmit the article.

Reviewer 3 ·

Basic reporting

no comment

Experimental design

no comment

Validity of the findings

no comment

Additional comments

It is much improved from the previous version.

Cite this review as

Reviewer 4 ·

Basic reporting

1. In the file named "code.py", it could not be seen how the data balancing operations mentioned in the "4.4 Experimental design" section were carried out.

2. Within the scope of the statement "using genetic algorithm to determine the network structure parameters training model of CNN" in the "4.2 DDoS attack detection system based on deep learning" section, there is no code section related to the genetic algorithm in the file named "code.py". In the study, no example of the detection of attack traffic using CNN was seen in the later stages.

3. Specifying the hyperparameters of ICWGANInverter used in the study in detail and presenting them in the code section will be explanatory for the details of the method discussed.

4. The coding for the statement "combining CWGAN-GP and WGANInverter, this paper proposes an ICWGANInverter" in the "4.3 Attack detection method of incomplete samples based on GAN" section cannot be seen in the file named "code.py".

5. The code content for the "Number of units in each layer of discriminator (40,20,10)" section in the "4.5 Experimental environment and parameter setting" section could not be seen in the file named "code.py".

6. The performance of the study should be evaluated separately for Normal, DoS, User to Root (U2R), Remote to Local (R2L) and Probe. Confusion matrix should be created to better evaluate the results.

7. It should be emphasized that this study differs from other studies in the literature that were tested using the KDDTest+ and KDDTest-21 datasets and showed much higher attack detection performance.

8. There was no consistency between the code in the file named "code.py" and the methods described in the study.

Experimental design

Within the scope of the statement "using genetic algorithm to determine the network structure parameters training model of CNN" in the "4.2 DDoS attack detection system based on deep learning" section, there is no code section related to the genetic algorithm in the file named "code.py". In the study, no example of the detection of attack traffic using CNN was seen in the later stages.

Validity of the findings

The performance of the study should be evaluated separately for Normal, DoS, User to Root (U2R), Remote to Local (R2L) and Probe. Confusion matrix should be created to better evaluate the results.

Annotated reviews are not available for download in order to protect the identity of reviewers who chose to remain anonymous.
Cite this review as

---

## Round 0.3 · Minor Revisions

Reviewer has suggested minor corrections. Authors are suggested to include them so that paper can be accepted.

Reviewer 4 ·

Basic reporting

The hyperparameters and their corresponding values used for the ICWGANInverter model in the study are not visible in the content of the article.

The section 5.2 could be made more concise by directly presenting the outcome without detailed computations.

Experimental design

no comment

Validity of the findings

no comment

Additional comments

It has been significantly enhanced compared to the previous version.

Cite this review as

---

## Round 0.4 · accepted · Accept

The paper can be accepted now.